# Exploring the Link Between Physical Activity, Sports Participation, and Loneliness in Adolescents Before and Into the COVID-19 Pandemic: The HUNT Study, Norway

**DOI:** 10.3390/ijerph21111417

**Published:** 2024-10-25

**Authors:** Vegar Rangul, Erik Reidar Sund, Jo Magne Ingul, Tormod Rimehaug, Kristine Pape, Kirsti Kvaløy

**Affiliations:** 1HUNT Research Centre, Department of Public Health and Nursing, Faculty of Medicine and Health Sciences, Norwegian University of Science and Technology (NTNU), 7600 Levanger, Norway; erik.r.sund@ntnu.no (E.R.S.); kirsti.kvaloy@ntnu.no (K.K.); 2Levanger Hospital, Nord-Trøndelag Hospital Trust, 7600 Levanger, Norway; 3Regional Centre for Child and Youth Mental Health and Child Welfare, Department of Mental Health, Faculty of Medicine and Health Sciences, Norwegian University of Science and Technology (NTNU), 7491 Trondheim, Norway; jo.m.ingul@ntnu.no (J.M.I.); tormod.rimehaug@ntnu.no (T.R.); 4Department of Public Health and Nursing, Faculty and Medicine and Health Sciences, Norwegian University of Science and Technology (NTNU), 7491 Trondheim, Norway; kristine.pape@ntnu.no; 5Centre for Sami Health Research, Department of Community Medicine, Faculty of UiT, The Arctic University of Norway, 9037 Tromsø, Norway

**Keywords:** the Trøndelag Health Study, physical activity, sports participation, loneliness, epidemiology

## Abstract

Background: The COVID-19 pandemic’s effects on adolescents’ physical activity, sports involvement, and feelings of loneliness remain inadequately understood. This study aimed to explore the shifts in leisure-time physical activity, sports participation, and loneliness among adolescents before and during the pandemic, positing that the pandemic has led to decreased physical activity and sports engagement, as well as heightened loneliness, where more active adolescents experience lower loneliness levels. This study included a prior four-year follow-up cohort from the same region two decades earlier to explore the existence of typical longitudinal aging effects in a cohort not affected by the pandemic. Methods: Prospective and longitudinal data from two cohorts of the Young-HUNT Study two decades apart involving adolescents aged 13–19 years were utilized. The controls were as follows: Cohort 1 from the Young-HUNT1 (YH1) Survey included 2399 adolescents with follow-up in the Young-HUNT2 (YH2) Survey four years later. Cohort 2 included the Young-HUNT4 (YH4) Survey (2017–2019) of 8066 adolescents, with a subset of 1565 participants followed up in the Young-HUNT COVID Survey (YHC) (2021) after exposure to the COVID-19 pandemic and associated restrictions. Changes over time were assessed using McNemar’s tests and dependent sample *T*-tests, while multinomial logistic regression modeled within-individual changes in loneliness, adjusting for age, gender, and other factors. Results: The findings revealed a significant decline in physical activity and sports participation in both cohorts from early to late adolescence. Additionally, there was a considerable increase in reported loneliness, more after exposure to the pandemic and especially among girls, but without any difference in historical initial levels (between cohorts). Inactive adolescents faced a greater risk of increased loneliness, while those participating in sports had a lower risk of loneliness. Physically inactive boys had a higher risk of loneliness compared with physically active boys at both time points in Cohort 2, which was higher than in the control Cohort 1. There was no historical difference between initial assessments. Conclusion: Adolescents experienced a significant decrease in physical activity and sports participation, along with increased loneliness, from early to late adolescence. Given the protective benefits of physical activity against loneliness and the negative longitudinal trends observed, public health initiatives should focus on increasing physical activity and reducing sports drop-out rates among adolescents to combat rising loneliness.

## 1. Introduction

Regular physical activity promotes good health and well-being throughout the course of life and is an effective strategy in the prevention of chronic disease and premature mortality [1,2]. Adolescence is an important period for the establishment of a healthy and active lifestyle. School and organized sports activities act to promote physical activity, but we know that many adolescents become physically inactive during adolescence [3]. Physical inactivity itself is considered a global epidemic [4], and it is estimated that 28% of the world population (1.4 billion people) is physically inactive [1]. In December 2019, we had the initial outbreak of the coronavirus disease (COVID-19); the virus spread rapidly globally [5], and as an attempt to control the virus dispersion, schools were closed, and commercial activities and non-essential services were suspended by the government institutions and health authorities in Norway and many other countries [6]. Physical distancing among people was considered fundamental to prevent the virus spread [7]. The implementation of social distancing was expected to result in a decline in everyday physical activity, hence negatively impacting daily physical activity in general [8,9]. Linked with pandemic-caused restrictions on leisure time activity commitment, one could anticipate a further decline in organized sports participation as well. This, and its potential impacts, are important to explore, considering the fact that overall physical activity levels generally diminish as age increases [10].

The ordered social distancing, lockdown of schools, and organized activities probably affected adolescents’ experience of loneliness. Adolescence is a particularly vulnerable period, where many experience loneliness more than later in life due to biological and psychosocial development and the social transition of adolescence [11,12]. Not being able to retain relationships with peers in this period of life may increase the risk of long-lasting loneliness. Independent of the pandemic period, loneliness has increased significantly from 2010 to 2017 in adolescents, particularly among girls [13]. This rise was also demonstrated in a recent Norwegian study where approximately twice as many girls as boys reported being very lonely [14]. Several studies have linked loneliness to risk factors such as low socioeconomic status and lack of social networks [15,16], and female sex is a significant predictor of loneliness [17]. The increase in loneliness also seems to be associated with the adverse mental health trend [14].

Physical activity can contribute to a decrease in loneliness when activity is organized in groups or shared environments, although a bidirectional relationship may also be present as loneliness itself might reduce the probability of being physically active [18]. A recent study on adults found that moderate and high physical activity levels were associated with reduced rates of severe loneliness [19]. Similar studies on adolescents are limited to cross-sectional studies [18].

One major consequence of the pandemic restrictions has been the negative impact on physical activity opportunities [20]. The organizational structure characteristics of many sports provide a unique opportunity for participants to engage both in health-enhancing physical activity and social interaction. A reduction in sports participation is assumed to reduce the social and mental health benefits facilitated by this participation [21]. A recent study suggested that physical activity is generally inversely related to feelings of loneliness. This relationship varied between pre-pandemic and during-pandemic assessments, with a significant relationship observed only in individuals assessed during the pandemic [22].

Reduced physical activity levels and increased loneliness during physical distancing periods, such as the COVID-19 pandemic, present great challenges for public health due to the increased risk and severity of mental health problems; hence, the benefits of infection safety come with a cost.

Previous cross-sectional research has shown a decline in physical activity levels among children and youth during the pandemic [23]. It is important to note that the decline in physical activity was not only due to the closure of facilities but also due to increased screen time and sedentary behavior as youth spent more time indoors. Efforts to increase physical activity post-pandemic should consider these impacts and the need for tailored interventions to encourage a return to active lifestyles for all age groups.

This study aimed to explore changes in leisure time, physical activity, sports participation, and loneliness from before and well into the COVID-19 pandemic. We hypothesized that during the pandemic, (i) physical activity and sports participation would decrease, (ii) the prevalence of loneliness would increase, and (iii) adolescents who are physically active and participate in sports would report lower levels of loneliness.

Since physical activity and participation in organized activities tends to decrease with age during adolescence, this could preclude the identification of declines in activities due to the pandemic and not merely due to increasing age in the longitudinal dataset prior- and into the pandemic period (Cohort 2). To distinguish these effects, a previous (1995–1997 to 2000–2001) 4-year follow-up cohort (Cohort 1) from the Trøndelag Health Study (The HUNT Study) was included for comparison.

## 2. Materials and Methods

### 2.1. Study Population and Design

The Young-HUNT Study is the adolescent part of the Trøndelag Health Study (The HUNT Study), a population-based health study inviting all inhabitants, aged 13 years and above, of primarily the northern part of Trøndelag County (former Nord-Trøndelag) in the central part of Norway [24]. The population was stable during the study period, homogenous and well-motivated to participate. Participants were invited by an invitation letter, which included information about this study and data security. The Young-HUNT Study consists primarily of several school-based surveys. Youth in the age range of 13–19 year olds were invited to participate during school hours. Young-HUNT1 (YH1) had a response rate of 88%. Those aged 13–16 in YH1 were invited to a follow-up study, Young-HUNT2 (YH2), four years later, with a response rate of 77% [25]. A flow chart for the design and number of participants in this study is shown in Figure 1. The YH1-YH2 cohort (here called Cohort 1) consisted of 2399 participants.

In the Young-HUNT4 (YH4) Survey, 2017–2019, 8066 (76%) adolescents between 13–19 years old participated, and those who were 13–16 years old were invited to the Young-HUNT COVID Survey (YHC) four years later. The YH4-YHC cohort (here called Cohort 2) consisted of 1565 participants.

This study was approved by The Norwegian University of Science and Technology (NTNU) based on a Data Protection Impact Assessment (DPIA). Ethical approval for this study was obtained from Regional Committee for Medical and Health Research Ethics (REC) in Norway, license number: 2021/381487. The HUNT4 data collection was mandated by a health registry and was, therefore, outside REC’s mandate for research ethics approvals. However, REC has offered feedback on the setup of the data collection and the consent form.

### 2.2. Measures

#### 2.2.1. Physical Activity

One of the exposures in this study was (reduced) physical activity in leisure time, assessed in both cohorts using a question on the duration of PA from the WHO Health Behavior in Schoolchildren Survey questionnaire (HEVAS/WHO HBSC) [26], which had been previously validated in the present population [27]. The question asks for the duration of time spent being physically active in sports and/or exercising outside school hours. The answers were dichotomized into “inactive” (<150 min/week) and “active” (≥150 min/week).

#### 2.2.2. Sports Participation

The second exposure was sports participation, where categorization of the responses was necessary to create identical dichotomous categories of “Yes” or “No” regarding “Participation in sport:

In YH1 sports participation was assessed by two questions regarding activity and type of sport: ‘Do you participate actively in sport? Those who answered “yes” or “No, but did before”, and indicating one of the 14 sport types, were classified as “participation in sport”.

In YH4, participants reported on the type and frequency of sport activity by the question, “How often do you participate in the following sports?”: endurance sports (e.g., cross-country skiing, swimming, running); team sports (e.g., soccer, volleyball, handball, hockey); aesthetic sports (e.g., dance, gymnastics); martial arts/strength sports (e.g., judo, karate, boxing, weightlifting); strength sport/body-building/fitness, technical sports (e.g., track and field, riding, ski-jump); skiing (e.g., alpine, snowboard, telemark); outdoor recreation (e.g., hiking, skiing); and fitness center training. Four alternatives were given for describing the frequency of participation in each of the sports categories: “never”, “less than once a week”, “once a week”, and “several times a week”. A frequency of “at least once a week” and at least one of the eight sport types were defined as “participation in sport”.

#### 2.2.3. Loneliness

Loneliness was the outcome, and a single question to assess loneliness was used in all the Young-HUNT surveys. The Young-HUNT1 and Young-HUNT2 item “Do you often feel lonely?” used a five-point response scale including “very often”, “often”, “sometimes”, “very rarely”, and “never”. In Young-HUNT4 and COVID, loneliness was assessed by the item, “At School or in leisure time, how often have you felt lonely?” with the same five-point response scale as in YH1 and YH2. For the main analysis, the response was dichotomized into “Not lonely”, including the responses “never” and “very rarely”; the other responses were classified as “Lonely”.

Within-individual changes in loneliness for the two time spans (YH1-YH2 or YH4-YHC) were categorized as “not lonely”, “increasing loneliness”, “lonely”, and “decreasing loneliness”, and the following four patterns were constructed: (1) not lonely group (not lonely in YH1/YH4 and YH2/YHC), (2) increasing loneliness group (not lonely YH1/YH4 and lonely YH2YHC), (3) decreasing loneliness group (lonely YH1/YH4 and not lonely YH2/YHC), and the lonely group (lonely in YH1/YH4/and YH2/YHC).

#### 2.2.4. Sociodemographic Factors

Sociodemographic characteristics of gender and age, socioeconomic status (SES), number of close friends, and academic aspirations were used as covariates and adjusted for in the logistic regression models for each longitudinal Young-HUNT dataset.

Educational aspirations were assessed by the item “What plans do you have regarding continued studies?”. The question had seven response alternatives: “college or university for 4 years or more”, “college or university <4 year”, “Secondary school, general subjects”, “Secondary school, vocational subjects”, “other vocational training”, “no plans” and “don’t know”. For the analyses, we collapsed it into three categories: the first was merged to “college or university”, the second category was those who reported “secondary school or other vocational training”, whereas the third category consisted of those who answered “no plans” and “don’t know” combined to “no plans/don’t know.

According to previous studies, friendships are a critical resource against loneliness, and not having any friends in adolescence was assumed to be comparable to a high loneliness status [28]. The occurrence of visiting friends was identified by the question, “how often are you together with friends?”. Of the five response items, adolescents reporting “never” were classified as never together with friends.

#### 2.2.5. Mental Distress

Mental distress was measured using the five-item Hopkins Symptom Checklist (HSCL-5). The adolescents were asked if they had experienced each of the following during the last 14 days: “Been constantly afraid and anxious”, “Felt tense or uneasy”, “Felt hopelessness about the future”, “Felt dejected or sad”, and “Worried too much about various things”. Each item was answered on a four-point scale: “Not at all”, “A little”, “Quite a bit”, and “Very much”. A mean HSCL score was calculated across the five items, with a cut-off score of ≥2 (symptoms of mental distress) [29]. Mental distress can affect both physical activity levels and loneliness and, therefore, operate as a mediator and confounder. We included mental distress as a confounder in some additional analyses (Appendix A) to ensure that any observed relationship between physical activity and loneliness is not simply due to underlying mental health conditions.

#### 2.2.6. Self-Rated Health

Self-rated health “at the moment” was assessed by a single question with a four-point labeled scale, with the following possible responses: “Poor”, “Not very good”, “Good”, and “Very good”. These responses were dichotomized as “not so good” included responses “Poor”, “Not very good”, and “good” included responses “Good” and “Very good”. This variable reflects an individual’s overall perception of their health, which can influence both their level of physical activity and their feelings of loneliness. By controlling for initial self-rated health, this study can better isolate the specific impact of physical activity on loneliness, independent of general health perceptions.

Self-rated health could also operate as a mediator and confounder, but by accounting for these possible confounders in the additional analysis, this study can more accurately determine the true relationship between physical activity and loneliness, ensuring that the results are not biased by these influential factors.

### 2.3. Statistical Analysis

Descriptive data are presented as percentages for categorical variables and means with standard deviations for continuous variables. Differences between baseline and follow-up were tested using McNemars’s tests and dependent (paired) sample *T*-tests. Within-individual change in loneliness was analyzed using Multinominal logistic regression with the four patterns of loneliness change between time points as a four-category dependent variable. A 95% confidence interval (CI) or *p* < 0.05 as a threshold for statistical significance was used in the analyses.

Each exposure was examined separately, only adjusted at baseline for age and gender (model 1), and then covariates (age, educational aspirations, visiting friends occurrence and gender) were included in model 2, additionally a fully adjusted model (model 3) in Appendix A. The choice of reference group in the analysis is based on the aim being studied. Missing values were handled using available case analysis, where each analysis included the cases with available data on the variables included in the analysis. The IBM SPSS Statistics software, version 29, was used for the statistical analyses.

## 3. Results

The study samples were the combination of two longitudinal cohorts: 2399 participants (girls *n* = 1284, 53.5%) in the Cohort 1 set, and 1565 participants (girls *n* = 960, 64.3%) in the Cohort 2 (Table 1). The mean age in YH1 was 14.5 (SD 0.9) and 18.3 (SD 0.8) in YH2 and, correspondingly, 14.6 (SD 1.0) in YH4 and 17.8 (SD 0.9) in YHC. For both cohorts, only those participating at both time points were included in our study sample.

Descriptive information regarding physical activity (PA) and sports participation in both cohorts is provided in Table 1. There is a significant longitudinal decrease in physical activity (min/week) in both cohorts, which is greater in Cohort 2 than in Cohort 1. Among girls, the average physical activity decreased from 177 min/week to 164 min/week in Cohort 1, while in Cohort 2, the PA decreased from 228 min/week to 172 min/week. Among boys, the same trend was observed, where PA decreased from 214 min/week to 203 min/week (Cohort 1) and from 249 min/week to 218 min/week (Cohort 2). Regarding the proportion of adolescents who participated in sports once or more a week, similar trends were revealed, where a substantial proportion stopped participating in sports activities during adolescence. While approximately 64% participated at age 13–15 years in YH1 (girls; 63.2%, boys; 65.5%) and above 70% (girls; 71.4%, boys; 72.8%) in YH4, this was reduced to below 40% (girls; 31.5%, boys; 38.2%) four years later in YH2 and YHC (girls; 34.1%, boys; 39.6%). The trend is similar but with differences at baseline (YH1 and YH4).

### 3.1. Prevalence of Loneliness

In general, more girls reported loneliness than boys. In YH1, 365 girls (29%) reported that they felt lonely “very often”, “often”, or “sometimes”, while in YH2 (aged 16–19 years old), this had increased to 423 (33.6%). In YH4, 261 girls (28.4%) reported feeling lonely, with an increased proportion to 421 (45.8%) in YHC (Table 1).

### 3.2. Loneliness Trends

Twice as many girls as boys were classified as “lonely” in both cohorts, where 16.2% of girls (8.9% of boys) were lonely in YH1 through YH2 and 18% (boys 7.5%) from YH4 through YHC. Overall, most participants reported not being lonely through adolescence from age 13–15 to age 16–19 (53.6% of girls and 67.8% of boys) in Cohort 1 (Figure 2). Among girls, this loneliness proportion was significantly lower in Cohort 2 (43.9%) compared with Cohort 1 (53.6%) compared with the stability seen in Cohort 1 (67.8%) compared with Cohort 2 among boys (66.1%).

The opposite was the case regarding the proportion of girls classified as “increasing loneliness” through adolescences, starting at 17.4% in Cohort 1 and increasing to 27.7% in Cohort 2—after exposure to the pandemic.

### 3.3. Physical Activity, Sports Participation, and Loneliness in Cohort 1

Table 2 shows associations between physical activity and sports participation with loneliness in Cohort 1. In model 2, there was a higher risk among the physically inactive (<150 min/week) for increasing loneliness across follow-up among both girls (OR 1.45, 95% CI 1.03–2.03) and boys (OR 1.70, 95% CI 1.12–2.58), and this is still valid in fully adjusted model 3 (Appendix A). A somewhat lower prevalence of loneliness (at both time points) was observed among girls (OR 0.58, 95% CI 0.42–0.81) and boys (OR 0.52, 95% CI 0.34–0.80) who participated in sports, also in the adjusted model. Both girls and boys who participated in sports had a lowered risk of loneliness across follow-up (OR 0.59, 95% CI 0.41–0.85, OR 0.68, 95% CI 0.46–0.99, respectively).

### 3.4. Physical Activity, Sports Participation, and Loneliness from Cohort 2

Association between physical activity and stable loneliness was observed among boys from YH4 through YHC (Cohort 2), with physically inactive boys (OR 4.27, 95% CI 2.19–10.22) and increasing loneliness from the first to the second time point (OR 2.91, 95% CI 1.61–5.26) (Table 3) compared with physically active boys.

Physical inactive girls were more likely to report loneliness in Cohort 2 (OR 2.26, 95% CI 1.37–3.72) compared with girls reporting physical activity. Boys who reported participation in sports had a lower risk of reporting loneliness at both time points (OR 0.40, 95% CI 0.16–0.95).

## 4. Discussion

This study compared changes in physical activity and loneliness over four years from early to late adolescence in two time periods, 1995–1997 to 2000–2001 (Cohort 1) and 2006–2008 to 2021 (Cohort 2), with the latter including the changes during the COVID-19 pandemic period. These were the main results:

Firstly, physical activity decreased with increasing age during adolescence in both genders, whereas loneliness increased only somewhat. Historically, the levels of physical activity in early adolescence increased considerably across the two decades between the first and the second period. When Cohort 2 reached late adolescence, approximately 1 year into the pandemic (spring 2021), their previous higher level of physical activity significantly decreased.

Secondly, an extensive decline in participation in organized sports from the age of 13–15 to the age of 16–19 was observed in both cohorts.

Thirdly, loneliness is more pronounced among girls in both cohorts, and the proportion that report loneliness increases with age in both sexes. However, the prevalence increased significantly more during the pandemic period than in the previous period two decades earlier.

Fourthly, physical inactivity was shown to be associated with loneliness in both cohorts, hence, independent of the pandemic.

Fifthly, participation in sports seems to be associated with a decrease in loneliness in both time periods among girls.

Several observational studies around the world have reported high and increasing proportions of youths being insufficiently physically active [1,3]. Globally, this comprised 81% of adolescents (11–17 years old) in 2016, although unequally distributed globally and within societies [30]. This fact, in addition to physical activity declining with increasing age (more than 25% decrease from age 9 to 15 years) [31], calls for the urgency of improving strategies to enhance physical activity levels in the adolescent population. Similar to many other studies, we also saw a slightly higher level of physical activity in our study comparing 2019 to 1997. Even so, the decrease in physical activity with increasing age is substantial, regardless of whether it was in the late 1990s or today. Analyzing our combined study sample showed that the COVID-19 pandemic was associated with a decrease in physical activity compared with three years earlier. A decline in physical activity was expected as lockdowns required the suspension of group-based sports and leisure facilities, and in many municipalities, outdoor-joint activities were restricted in general. This probably led to general difficulties in maintaining regular physical activity routines during lockdown. This is in line with findings in other studies [20], but as pointed out, this finding could reflect only typical changes with age and not the COVID-19 pandemic. In our study, we could show a marked difference by using a comparable longitudinal cohort (1995–1997 to 2000–2001) as a control for the potential changes due to the pandemic. Another obvious reason for the decline from early to late adolescence is the actual higher age in the latter and what that brings with it concerning higher drop-out from organized physical activity participation in general.

The organizational structure of sports provides a unique opportunity for adolescents to engage in health-enhancing physical activity and, at the same time, socialize with other peers. Sport and exercise are beneficially associated with a wide range of advantages that extend beyond physical health, including psychological health outcomes and social gains, which further positively affect various aspects of life [21,32,33]. Participation in organized sports suffers from a huge drop-out over time from early to late adolescence, and we saw in our study that from the age 13–15 to 16–19, the proportion who participate is halved. In addition, many have already dropped out before they are 13 years old.

Interestingly, our data cannot reveal any substantially higher drop-outs in the pandemic period compared with the previous period that was studied, which was surprising. We could hypothesize that the degree of closure in our region, which is mostly rural, was not as affected by the restrictions as could be assumed by getting information from other more urban parts of Norway.

Very few studies have examined the relationship between sports participation and loneliness. However, several studies have focused on this indirectly, for instance, investigating whether participating in sports is beneficial for social outcomes, such as self-control, interpersonal communication, and a sense of belonging [34,35]. Loneliness is both a cause and consequence of poor mental health and social outcomes [36], and even when in our study we adjusted for potential confounding factors, it seems that participating in sports was associated with a lower risk of loneliness, especially among boys. A similar association was found with physical activity, where boys who were physically active had a lower risk of loneliness. One can assume that a decline in participation in sports can lead to a decrease in daily physical activity and an increase in loneliness. This is in line with the literature, which states that when boys participate in sports, they often experience a sense of belonging and social connection [37], which can significantly reduce feelings of loneliness. Conversely, a decline in sports participation can lead to decreased daily physical activity, which may reinforce loneliness [38].

It is important to note that Norwegian municipalities differed in the severity of restrictions and lockdowns within the region across different dates, making it difficult to objectively assess how this may have impacted sports opportunities and, hence, behaviors. For example, in areas where outdoor exercise was permitted with social distancing, people might have maintained or even increased their physical activity levels. In contrast, those in stricter lockdowns without the ability to leave home might have experienced a significant reduction in their physical activity, potentially affecting their overall health and well-being. This variability makes it challenging to assess the overall impact of lockdowns on physical activity and sports opportunities. It also highlights the importance of considering local context when evaluating public health measures and their effects.

This study has several strengths and some limitations. A major strength of this study is the longitudinal design from an unselected general population, allowing us to examine the same adolescents at two time points and assess the associations between physical activity and sports participation with loneliness. In addition, we have two datasets, one being collected two decades prior to the pandemic, which gives the opportunity to generalize regarding the effects that are revealed. The sampling within the Young-HUNT Study was conducted using the same methods and instruments as all surveys included in the present study. The population is stable and homogenous. To our knowledge, this is one of the very few population-based health studies with such comprehensive information about sports participation in relation to age and gender, enabling us to examine the longitudinal associations between sports participation and loneliness. Because loneliness is both a cause and consequence of mental health problems, another strength is that we adjusted for mental health problems (Hopkins Symptom Checklist, HSCL-5) that may otherwise have affected both physical activity, sports participation, and loneliness.

The main limitation of this study is the data being self-reported and, therefore, being susceptible to information bias. We used single-item measure to assess physical activity and loneliness, and the variables to describe sports participation exposure provided a crude measure of participation frequency. However, the WHO HBSC questions for physical activity used in this study have been shown to hold acceptable validity and reliability in adolescents [27]. The single loneliness measure has been shown to have lower reliability and validity than multi-item scales, avoiding the term “lonely” [39]. A low threshold was used to capture and include all severity levels of loneliness, with the recognition that mild levels can also increase the risk of mental and physical health outcomes. We should also address potential selection biases that may affect the generalizability of the findings. Participants who chose to engage in physical activity and sports participation may differ systematically from those who did not, potentially biasing the results. Second, non-identical measurements may introduce measurement errors. Differences in data collection methods or participant rates between the different time points may affect the accuracy of the observed changes.

Although the response rates in the Young-HUNT1 and Young-HUNT4 were high, the lower response rate among upper secondary school students (8th–10th grade/13–15 years old) compared with lower secondary school students (11th–13th grade/16–19 years old)) may represent a selection bias.

Finally, residual confounding is still likely due to unmeasured or insufficiently measured confounders.

## 5. Conclusions

During adolescence, independent of the COVID-19 lockdown, physical activity levels and sports participation decreased remarkably, while loneliness increased correspondingly, especially among girls linked to the pandemic, which adds to the understanding of how these activities impact boys and girls differently. Considering the negative trend and impacts of loneliness and the protective role of physical activity against loneliness during adolescence in both genders, with sports participation linked to reduced loneliness, it is advised that public health initiatives focus on boosting physical activity and minimizing sports drop-out rates among adolescents. A special focus should be on monitoring post-lock-down situations; however, the biggest challenge is linked to the impact of drop-out due to increasing age, and a much greater focus must be placed on preventing this in adolescence. The added emphasis on monitoring post-lockdown situations and addressing the challenges of age-related drop-out provides a more comprehensive approach to sustaining adolescent engagement in physical activities. Sports initiatives must have a lower threshold for participation during adolescence, emphasizing development and enjoyment rather than competition, which is often driven by parents’ ambitions.

## Figures and Tables

**Figure 1 ijerph-21-01417-f001:**
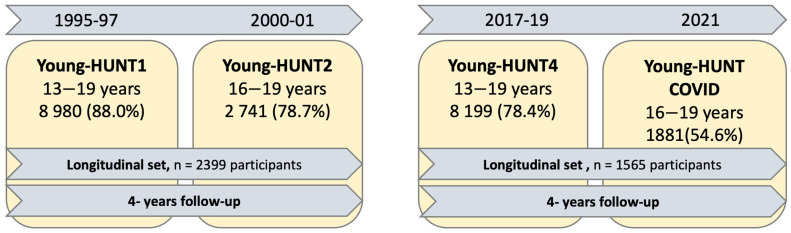
Flowchart of the Young-HUNT cohorts and participation in the survey waves: participants’ age ranges, sample size and response rate (each longitudinal data set = analytical cohort 1/2).

**Figure 2 ijerph-21-01417-f002:**
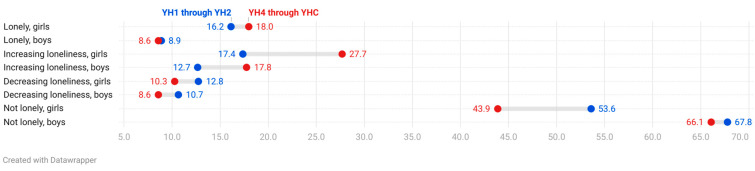
Comparing cohorts regarding the proportions (%) of self-reported loneliness from age 13–15 (YH1 and YH4) to age 16–19 (YH2 and YHC) in two cohorts studied two decades apart. YH1 through YH2 (1995–2000) and YH4 through YHC (2017–2021).

**Table 1 ijerph-21-01417-t001:** Descriptive characteristics of the adolescents by study, stratified by sex.

		Young-HUNT1		Young-HUNT2			Young-HUNT4		Young-HUNT COVID		
		*n*	%	*n*	%	*p* ^#^	*n*	%	*n*	%	*p* ^#^
Age, mean ± SD											
	Girls	1284 *	14.6 ± 0.9		18.4 ± 0.8	<0.001	960	14.7 ± 1.0		17.8 ± 0.9	<0.001
	Boys	1115 *	14.5 ± 0.9		18.3 ± 0.8	<0.001	605	14.6 ± 1.0		17.8 ± 0.9	<0.001
Physical activity (min/week), mean ± SD											
	Girls	1261	177 ± 122	1265	164 ± 130	<0.001	933	228 ± 141	947	172 ± 145	<0.001
	Boys	1097	214 ± 136	1096	203 ± 151	0.017	589	249 ± 142	600	218 ± 154	<0.001
Participate in sports (Yes, once or more week)											
	Girls	812	63.2	405	31.5	<0.001	673	71.4	322	34.1	<0.001
	Boys	730	65.5	426	38.2	<0.001	432	72.8	239	39.6	<0.001
Loneliness (feeling lonely)											
	Girls	365	29	423	33.6	0.003	261	28.4	421	45.8	<0.001
	Boys	213	19.6	235	21.6	0.188	92	16.1	102	25.3	<0.001
Educational aspirations											
	Girls, No Plans/don’t know	508	39		na		404	61.3		na	
	Boys, No Plans/don’t know	568	45.2		na		263	58.1		na	
	Girls, High School or other education	626	48.0		na		47	7.1		na	
	Boys, High School or other education	542	43.1		na		82	18.1		na	
	Girls, University or College	170	13.0		na		208	31.6		na	
	Boys, University or College	147	11.7		na		108	23.8		na	
Self-rated health (Not so good)											
	Girls	104	8.3	175	13.9	<0.001	111	11.6	184	19.3	<0.001
	Boys	79	7.2	112	10.3	0.006	63	10.5	69	11.5	<0.001
Mental distress, HSCL-5 (symptoms, ≥2.0)											
	Girls	164	13.4	329	27.0	<0.001	304	34.1	467	52.4	<0.001
	Boys	74	7.0	130	12.2	<0.001	82	15.0	124	22.6	<0.001
Together with friends (never)											
	Girls	84	6.7	64	5.1	0.00	35	4.0	53	6.0	0.054
	Boys	103	9.5	76	7.0	0.028	31	5.8	60	11.3	<0.001

* Subgroups may not total to this number due to missing values; ^#^ McNemar test for categorical variables and dependent (paired) sample *t*-test for continuous variables. na = data not available.

**Table 2 ijerph-21-01417-t002:** Odds ratios (OR) with 95% confidence intervals (CIs) for the probability of loneliness through four years from YH1 to YH2 (Cohort 1, *n* = 2066) measured by stable loneliness, increasing or decreasing loneliness compared with not being lonely predicted by physical activity and sports participation stratified on gender.

		Outcome					
		Model 1			Model 2		
		Lonely OR (95% CI)	Decreasing Loneliness OR (95% CI)	Increasing Loneliness OR (95% CI)	Lonely OR 95% CI	Decreasing Loneliness OR 95% CI	Increasing Loneliness OR 95% CI

Girls							
Physical activity							
	Active (≥150 min/wk)	1.0	1.0	1.0	1.0	1.0	1.0
	Inactive (<150 min/wk)	1.57 ** (1.11–2.21)	1.30 (0.89–1.90)	1.56 ** (1.12–2.17)	1.45 * (1.02–2.08)	1.29 (0.87–1.90)	1.45 * (1.03–2.03)
Boys							
Physical activity							
	Active (≥150 min/wk)	1.0	1.0	1.0	1.0	1.0	1.0
	Inactive (<150 min/wk)	1.55 (0.59–2.51)	1.39 (0.89–2.18)	1.87 ** (1.25–2.78)	1.34 (0.81–2.22)	1.38 (0.87–2.19)	1.70 ** (1.12–2.58)
Girls							
Sports participation							
	No	1.0	1.0	1.0	1.0	1.0	1.0
	Yes	0.58 *** (0.42–0.81)	0.57 *** (0.40–0.82)	0.79 (0.57–1.09)	0.61 ** (0.43–0.85)	0.59 * (0.41–0.85)	0.82 (0.59–1.13)
Boys							
Sports participation							
	No	1.0	1.0	1.0	1.0	1.0	1.0
	Yes	0.52 ** (0.34–0.80)	0.81 (0.53–1.22)	0.68 * (0.46–0.99)	0.54 * (0.34–0.85)	0.82 (0.54–1.26)	0.73 (0.49–1.08)

Model 1—Associations adjusted for age and stratified by sex in YH1. Model 2—Associations adjusted for age, educational aspirations, visiting friends occurrence and stratified by sex in YH1. ORs Odds Ratios, CI Confidence Interval, * *p* < 0.05, ** *p* < 0.01, *** *p* < 0.001. Not lonely = not lonely in YH1 and YH2, Lonely = lonely in YH1 and YH2, Decreasing loneliness = changing loneliness category from lonely in YH1 to not lonely in YH2, Increasing loneliness = not lonely in YH1 and lonely in YH2.

**Table 3 ijerph-21-01417-t003:** Odds ratios (ORs) with 95% confidence intervals (CIs) for the probability of loneliness through four years from YH4 to YHC (Cohort 1, *n* = 1565) measured by stable loneliness, increasing or decreasing loneliness compared with not being lonely predicted by physical activity and sports participation stratified on gender.

		Outcome					
		Model 1			Model 2		
		Lonely OR (95% CI)	Decreasing Loneliness OR (95% CI)	Increasing Loneliness OR (95% CI)	Lonely OR 95% CI	Decreasing Loneliness OR 95% CI	Increasing Loneliness OR 95% CI

Girls							
Physical activity							
	Active (≥150 min/wk)	1.0	1.0	1.0	1.0	1.0	1.0
	Inactive (<150 min/wk)	2.05 *** (1.35–3.08)	0.92 (0.52–1.63)	1.32 (0.91–1.94)	2.26 * (1.37–3.72)	1.09 (0.55–2.14)	1.50 (0.95–2.38)
Boys							
Physical activity							
	Active (≥150 min/wk)	1.0	1.0	1.0	1.0	1.0	1.0
	Inactive (<150 min/wk)	4.06 *** (2.07–7.97)	2.39 * (1.08–4.20)	2.21 ** (1.32–3.68)	4.74 *** (2.19–10.22)	2.11 (0.96–4.62)	2.91 *** (1.61–5.26)
Girls							
Sports participation							
	No	1.0	1.0	1.0	1.0	1.0	1.0
	Yes	0.48 *** (0.32–0.72)	0.99 (0.58–1.71)	0.68 (0.47–0.98)	0.85 (0.48–1.49)	1.33 (0.66–2.66)	0.70 (0.44–1.12)
Boys							
Sports participation							
	No	1.0	1.0	1.0	1.0	1.0	1.0
	Yes	0.28 *** (0.14–0.54)	0.80 (0.40–1.60)	0.60 * (0.37–0.98)	0.40 * (0.16–0.95)	1.06 (0.44–2.53)	0.60 (0.32–1.10)

Model 1—Associations adjusted for age in YH1 and sex. Model 2—Associations adjusted for age, educational aspirations, visiting friends occurrence and stratified by sex in YH1. ORs Odds Ratios, CI Confidence Interval, * *p* < 0.05, ** *p* < 0.01, *** *p* < 0.001. Not lonely/Lonely = self-reported lonely in YH4 and YHC, Decreasing loneliness = lonely YH4 and not lonely YHC, Increasing loneliness = not lonely YH4 and lonely YHC.

## Data Availability

Due to restrictions imposed by the HUNT Research Centre, in accordance with the Norwegian Data Inspectorate’s guidelines, data cannot be made publicly available. The data is currently stored in the HUNT Databank, and there are restrictions for handling data files. Data may be available upon request to the HUNT Data Access Committee (hunt@medicine.ntnu.no). The HUNT data access information (available at http://www.ntnu.edu/hunt/data. accessed on 21 October 2024) describes in detail the policy regarding data availability.

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
