# Peer review of "Exploring the Link Between Physical Activity, Sports Participation, and Loneliness in Adolescents Before and Into the COVID-19 Pandemic: The HUNT Study, Norway"

_ijerph, 2024, doi:10.3390/ijerph21111417_

Round 1
Reviewer 1 Report
Comments and Suggestions for Authors
Review of the manuscript: Exploring the Link Between Physical Activity, Sport Participa-2 tion, and Loneliness in Adolescents Before and Into the 3 COVID-19 Pandemic: The HUNT Study, Norway
The authors analyzed two longitudinal dataset from the larger HUNT-study from before and during the COVID-19 pandemic. The main findings were that physical activity and sports participation decreased during the adolescent years while loneliness rose, and that these changes were more pronounced during the pandemic than before. I have some recommendations to the authors on how to strengthen the communication of the findings.
1. The Introduction provides justification for the aims of the study, although the text appears a bit messy. I recommend that you revise the text so it appears shorter and more focused. Consider to delete the texts about mental health as this is not part of the study. The text after the aim (line 99-103) is difficult to understand if you are not familiar with HUNT.
2. The Methods sectiuon includes most of what the reader needs to understand the study. I recommend some revisions:
I guess that everything makes sense if you have a detailed insight into HUNT but for outsiders, some of the text is difficult to grasp. Some examples: Line 118 says ”In the Young-HUNT4 (YH4) Survey, 2017-19, 8066 (76%)” and I ask: 76% of what? Lnje 117 says ”consisted of 2399 participants from” and I ask: from what?
Figure 1 presents study populations that are irrelevant for this manuscript. This was confusing to me and I recommend that you revise the Figure so the reader understands which study populations are relevant in this particular study.
Consider to provide a declaration on which variables are exposures and which outcomes.
In some instances, the measurements are not similar at baseline and follow-up. Please address this issue in the Discussion section.
Variables such as sex, age, socioeconomic background are standard control variables and need no further justification. But I do recommend that you justify inclusion of self-rated health, mental distress, number of close friends, and academic aspirations in the analyses. There is no mentioning of these issues in the Introduction section it is difficult to understand why these variables, out of the rich dataset, were included in the analyses.
The statistical analyses separate a Model 1 which adjusts for age and gender and a Model 2 which adjusts for a long range of variables. Some of these variables may not be confounders but rather explanatory variables (mediators). If they are explanatory variables, for instance if they timewise appear after the exposure variable, then it is inappropriate to adjust for them. One example: Some of the OR estimates in Table 2 change a lot from Model 1 to Model 2. Is this due to confounding or mediation? I strongly recommend to remove the above mentioned variables from the analyses, alternatively to provide a solid justification for their inclusion.
3. I have a few recommendations for revisions of the Results section.
There may be a need to revise Table 1 and make sure that all figures are correct and understandable. One example: The table explains that 365 girls were lonely corresponding to 29.0% But 365 out of 1284 girls is 28.4%.
The column-heading “%” includes both percentages and minutes per week. The column-heading “p-trend” is inappropriate if it refuse of use of chi2-test. The chi2-test is a test for homogeneity, not for trend.
Figure 2 confused me. I thought that the sum of the four outcome-categories should be 100% but that is not the case. It may be better to substitute the Figure with a table.
4. The Discussion includes most of what you expect in this kind of a social-epidemiological paper. There is a nice highlight of main findings, comparison with relevant literature, and interesting attempts to interpret the findings. There is an oversight over strengths and limitations. I suggest that you expand the limitations to discuss potential selection bias and problems related to non-identical measurements at baseline and follow-up.
There is a fine text about the study’s practical implications but I would like the authors also to mention what they think are important implications for research.
Author Response
Response to Reviewer 1 Comments
Thank you very much for taking the time to review this manuscript. Please find the detailed responses below and the corresponding revisions/corrections highlighted/in track changes in the re-submitted files.
Comment from reviewer: The authors analyzed two longitudinal dataset from the larger HUNT-study from before and during the COVID-19 pandemic. The main findings were that physical activity and sports participation decreased during the adolescent years while loneliness rose, and that these changes were more pronounced during the pandemic than before. I have some recommendations to the authors on how to strengthen the communication of the findings.
Response: Thank you very much for taking the time to review this manuscript. Please find the detailed responses below and the corresponding revisions/corrections highlighted/in track changes in the re-submitted files.
Comment 1: The Introduction provides justification for the aims of the study, although the text appears a bit messy. I recommend that you revise the text so it appears shorter and more focused. Consider to delete the texts about mental health as this is not part of the study. The text after the aim (line 99-103) is difficult to understand if you are not familiar with HUNT.
Response 1: Thank you. Have revised some text, also the text after the aim, because this is an essential aspect of the study. We also carried out a thorough review of the text and language in the manuscript.
Page 2, line 58-64: The implementation of social distancing was expected to result in everyday physical activity decline and hence negatively impacting daily physical activity in general [8,9]. Linked with pandemic-caused restrictions on leisure time activity commitment one could anticipate a further decline also in organized sports participation. This, and the potential impacts of it is important to explore considering the fact that the overall physical activity levels did generally diminish as age increases
Page 3, line 106-111: Since physical activity and participation in organized activities tends to decrease with age during adolescence, this could preclude the identification of declines in activities due to the pandemic and not merely due to increasing age in the longitudinal dataset prior- and into the pandemic period (Cohort 2). To distinguish these effects, a previous (1995-97 to 2000-01) 4-year follow-up cohort (Cohort 1) from the Trøndelag health Study (The HUNT Study) was included for comparison.
Comment 2: The Methods sectiuon includes most of what the reader needs to understand the study. I recommend some revisions:
I guess that everything makes sense if you have a detailed insight into HUNT but for outsiders, some of the text is difficult to grasp. Some examples: Line 118 says ”In the Young-HUNT4 (YH4) Survey, 2017-19, 8066 (76%)” and I ask: 76% of what? Lnje 117 says ”consisted of 2399 participants from” and I ask: from what?
Figure 1 presents study populations that are irrelevant for this manuscript. This was confusing to me and I recommend that you revise the Figure so the reader understands which study populations are relevant in this particular study.
Consider to provide a declaration on which variables are exposures and which outcomes.
In some instances, the measurements are not similar at baseline and follow-up. Please address this issue in the Discussion section.
Variables such as sex, age, socioeconomic background are standard control variables and need no further justification. But I do recommend that you justify inclusion of self-rated health, mental distress, number of close friends, and academic aspirations in the analyses. There is no mentioning of these issues in the Introduction section it is difficult to understand why these variables, out of the rich dataset, were included in the analyses.
The statistical analyses separate a Model 1 which adjusts for age and gender and a Model 2 which adjusts for a long range of variables. Some of these variables may not be confounders but rather explanatory variables (mediators). If they are explanatory variables, for instance if they timewise appear after the exposure variable, then it is inappropriate to adjust for them. One example: Some of the OR estimates in Table 2 change a lot from Model 1 to Model 2. Is this due to confounding or mediation? I strongly recommend to remove the above mentioned variables from the analyses, alternatively to provide a solid justification for their inclusion.
Response 2: Thank you for these points in the methods section.
Figure 1, is not irrelevant regarding this study that relies on two longitudinal dataset, that includes four cross-sectional samples (YH1, YH2, YH4 and YHC). We think it is important to show numbers and participation rates in this cohorts and differentiate on both the longitudinal analytical sample and total cohort. We also added new text to figure 1: Flowchart of the Young-HUNT cohorts and participation in the survey waves: participants age ranges, sample size and response rate (each longitudinal data set = analytical cohort1/2).
We have included in the text what is exposure and outcome.
Page 4, line 143: One of the exposures in this study was Physical activity in leisure time, assessed by a question on duration of PA from…
Page 4, line 150: The second exposure was sport participation.
Page 4, line 168: Loneliness was the outcome, and a single question to assess loneliness was used in all the Young-HUNT surveys.
Our reason for including initial factors (self-rated health, mental distress, number of close friends, and academic aspirations) is that they could be potential confounders for our main results.
This is no elaborated in the text
Page 4-5, line 183-222:
Sociodemographic characteristics of gender and age, socioeconomic status (SES), number of close friends, and academic aspirations were used as covariates and adjusted for in the logistic regression models for each longitudinal Young-HUNT dataset.
Educational aspirations were assessed by the item “What plans do you have re-garding continued studies?”. The question had seven response alternatives: “college or university for 4 years or more”, “college or university <4 yr,”, “Secondary school, general subjects”, “Secondary school, vocational subjects”, “other vocational training,” “no plans” and “don’t know”. For the analyses we collapsed it into three categories: the first were merged to “college or university”, the second category was those who reported “sec-ondary school or other vocational training”, whereas the third category consisted of those who answered, “no plans” and “don’t know” combined to “no plans/don’t know.
According to previous studies, friendships are a critical resource against loneliness, and not having any friends in adolescence was assumed to be comparable to a high loneliness status [28]. The occurrence of visiting friends was identified by the question “how often you are together with friends?”. Five response items, were adolescents re-porting “never”, was classified as never together with friends.
2.2.5. Mental distress
Mental distress was measured using the 5-item Hopkins Symptom Checklist (HSCL-5). The adolescents were asked if they had experienced each of the following during the last 14 days: ‘Been constantly afraid and anxious’, ‘Felt tense or uneasy’, ‘Felt hopelessness about the future’, ‘Felt dejected or sad’, ‘Worried too much about various things’. Each item was answered on a four-point scale: “Not at all”, “A little”, “Quite a bit”, and “Very much”. A mean HSCL-score was calculated across the five items, with cut-off score of ≥ 2 (symptoms of mental distress) [29]. Mental distress can affect both physical activity levels and loneliness. Including mental distress as a confounder helps to ensure that any observed relationship between physical activity and loneliness is not simply due to underlying mental health conditions.
2.2.6. Self-rated health
Self-rated health ‘at the moment’ was assessed by a single question with a four-point labelled scale; with the following possible responses: “Poor”, “Not very good”, “Good”, and “Very good”. These responses were dichotomized as “not so good” included re-sponses “Poor”, “Not very good”, and “good” included responses “Good”, and “Very good”. This variable reflects an individual’s overall perception of their health, which can influence both their level of physical activity and their feelings of loneliness. By con-trolling for initial self-rated health, the study can better isolate the specific impact of physical activity on loneliness, independent of general health perceptions.
By accounting for these confounders, the study can more accurately determine the true relationship between physical activity and loneliness, ensuring that the results are not biased by these other influential factors
Response to comment on model 1 and model 2, same as above.
Comment 3: I have a few recommendations for revisions of the Results section.
There may be a need to revise Table 1 and make sure that all figures are correct and understandable. One example: The table explains that 365 girls were lonely corresponding to 29.0% But 365 out of 1284 girls is 28.4%.
The column-heading “%” includes both percentages and minutes per week. The column-heading “p-trend” is inappropriate if it refuse of use of chi2-test. The chi2-test is a test for homogeneity, not for trend.
Figure 2 confused me. I thought that the sum of the four outcome-categories should be 100% but that is not the case. It may be better to substitute the Figure with a table.
Response 3: Thank you.
Table 1. Due to missing values on some variables, there will be some differences in numbers comparing to girls/boys variable. Have included a clarification in the table.
Table 1: * Subgroups may not total to this number, due to missing values.
There were two wrong numbers in figure 2. Definitely, the sum should be 100%, and this is now corrected.
The column-heading “%” includes both, but for Age and physical activity variables we use continuous data, therefore, for these variables we indicate that we present age and minutes per week and standard deviation.
We agree that “p-trend” is inappropriate. We are comparing proportions between two related groups, and we have now used McNemar’s test to analyze paired nominal data to determine if there are differences between these two related groups (baseline and follow-up). These data are now interpreted in table 1.
Page 5, line 226-227: Differences between baseline and follow-up were tested using McNemars’s tests and dependent (paired) sample T-tests
Comment 4: The Discussion includes most of what you expect in this kind of a social-epidemiological paper. There is a nice highlight of main findings, comparison with relevant literature, and interesting attempts to interpret the findings. There is an oversight over strengths and limitations. I suggest that you expand the limitations to discuss potential selection bias and problems related to non-identical measurements at baseline and follow-up.
There is a fine text about the study’s practical implications but I would like the authors also to mention what they think are important implications for research
Response 4: Thank you. We have elaborated regarding potential limitations in the text.
Page 11, line 406-411: We should also address potential selection biases that may affect the generalizability of the findings. Participants who chose to engage in physical activity and sports participation may differ systematically from those who did not, potentially biasing the results. Second, non-identical measurements may introduce measurement error. Differences in data collection methods or participant rates between the different time points may affect the accuracy of the observed changes.
Reviewer 2 Report
Comments and Suggestions for Authors
Generally, this is a concise, and well-written manuscript.
· The authors clearly state what they intend to achieve with the present research. They are clear and specific.
· Introduction cites a fair amount of scientific research and their findings relevant to the topic.
· The methodology and data analysis are appreciated.
· Figure and Tables are appropriate. They show properly the data in details. They sufficiently present study's findings.
· The results are clear and well-written. span > /span/spanspan lang="EN-US" style="text-indent: -18pt"Authors successfully analyze the results of the research. They provide an appreciated interpretation of the results and the conclusions are supported by the data./span/p p class="MsoListParagraphCxSpMiddle" style="text-align: justify; text-indent: -18.0pt; mso-list: l0 level1 lfo1"Authors highlight the strengths of the study.
· Authors have clearly stated the limitations of the study.
The language of the paper is free from jargons using appropriate academic language.
Overall, it is a good quality manuscript. It fits to journal’s scope.
The truth is that I am not sure about the scope of the work’s appeal and its applications.
However, the results can be a good starting point for further research. It is a public health work with a focus on the sensitive group of teenagers with care for continued physical activity not only in Norway but also in other countries.
Based on the above comments the present paper could be accepted in the present form.
Author Response
Thank you very much for taking the time to review this manuscript. Please find the detailed responses below and the corresponding revisions/corrections highlighted/in track changes in the re-submitted files.
Comment summary: Generally, this is a concise, and well-written manuscript.
- The authors clearly state what they intend to achieve with the present research. They are clear and specific.
- Introduction cites a fair amount of scientific research and their findings relevant to the topic.
- The methodology and data analysis are appreciated.
- Figure and Tables are appropriate. They show properly the data in details. They sufficiently present study's findings.
- The results are clear and well-written. span > /span/spanspan lang="EN-US" style="text-indent: -18pt"Authors successfully analyze the results of the research. They provide an appreciated interpretation of the results and the conclusions are supported by the data./span/p p class="MsoListParagraphCxSpMiddle" style="text-align: justify; text-indent: -18.0pt; mso-list: l0 level1 lfo1"Authors highlight the strengths of the study.
- Authors have clearly stated the limitations of the study.
The language of the paper is free from jargons using appropriate academic language.
Overall, it is a good quality manuscript. It fits to journal’s scope.
The truth is that I am not sure about the scope of the work’s appeal and its applications.
However, the results can be a good starting point for further research. It is a public health work with a focus on the sensitive group of teenagers with care for continued physical activity not only in Norway but also in other countries.
Based on the above comments the present paper could be accepted in the present form.
Response: Thank you very much for taking the time to review this manuscript, and the very positive comments on this study. We have tried to elaborate the application/implications of these findings in the text.
Reviewer 3 Report
Comments and Suggestions for Authors
Dear authors.
I think your research is interesting enough to be published, however it deals with an outdated topic such as the COVID-19 pandemic. It is well known in the scientific literature that during the COVID-19 pandemic, physical activity levels decreased in all populations. I do not think the study adds anything new in this respect.
Continuing with the wording of the research, it is necessary that the text is written in the impersonal. The first person plural is observed (line 49). Revise the whole text to avoid this mistake.
The theoretical framework provides already known data. Why don't you contextualise through the theoretical framework the state of your issue in the area where you have carried out the study? This may help to better understand the results you later present. Only one research objective and three study hypotheses are presented. Propose an objective for each hypothesis. I believe that through this the research can be improved.
Moving on to the results. Some variables have not been collected with reliable and validated instruments (sport participation, loneliness). These variables cannot be considered reliable, as they have not been collected with instruments that have demonstrated a high degree of internal reliability. This is quite worrying and compromises the results obtained. Furthermore, no reliability data are shown for any of the instruments used.
Based on these results, I consider that the research cannot be published for the following two reasons:
-Lack of relevance of the subject matter addressed, as this has been extensively studied. This research does not contribute anything new.
-Lack of reliability of the results obtained. No tests have been carried out to show that the instruments used are reliable. Moreover, some instruments have not been validated. Most of these are self-report instruments.
Comments on the Quality of English LanguageThe wording of the research should be revised.
Author Response
Comment summary: Dear authors.
I think your research is interesting enough to be published, however it deals with an outdated topic such as the COVID-19 pandemic. It is well known in the scientific literature that during the COVID-19 pandemic, physical activity levels decreased in all populations. I do not think the study adds anything new in this respect.
Continuing with the wording of the research, it is necessary that the text is written in the impersonal. The first person plural is observed (line 49). Revise the whole text to avoid this mistake.
The theoretical framework provides already known data. Why don't you contextualise through the theoretical framework the state of your issue in the area where you have carried out the study? This may help to better understand the results you later present. Only one research objective and three study hypotheses are presented. Propose an objective for each hypothesis. I believe that through this the research can be improved.
Moving on to the results. Some variables have not been collected with reliable and validated instruments (sport participation, loneliness). These variables cannot be considered reliable, as they have not been collected with instruments that have demonstrated a high degree of internal reliability. This is quite worrying and compromises the results obtained. Furthermore, no reliability data are shown for any of the instruments used.
Based on these results, I consider that the research cannot be published for the following two reasons:
-Lack of relevance of the subject matter addressed, as this has been extensively studied. This research does not contribute anything new.
-Lack of reliability of the results obtained. No tests have been carried out to show that the instruments used are reliable. Moreover, some instruments have not been validated. Most of these are self-report instruments.
Response: Thank you for taking the time to review this manuscript and for your insightful comments on this study. Most studies conducted focus one aspects, either the pandemic, the age-trends, the associations between factors or uses cross-sectional designs. Our study combines longitudinal data with comparable cohorts and the pandemic exposures. This strong design combined with large representative samples is our contribution. By understanding these dynamics, policymakers, educators, and healthcare providers can develop effective strategies to support youth well-being, both during and beyond pandemic situations.
Regarding your concern for the reliability of the instruments and that they are self-reported: We claim that the requirement for individual and clinical use of instruments are not relevant in large-scale epidemiology. The questions used in this study are validated for epidemiological research and are widely accepted in the field. Such variables are common in epidemiological studies and continue to be relevant for understanding public health trends despite their limited scope. We agree that self-report instruments can introduce bias and affect the accuracy of the data. This weakness is already elaborated in strengths and limitations.
Page 11, line 406-411 :
We should also address potential selection biases that may affect the generalizability of the findings. Participants who chose to engage in physical activity and sports participation may differ systematically from those who did not, potentially biasing the results. Second, non-identical measurements may introduce measurement error. Differences in data collection methods or participant rates between the different time points may affect the accuracy of the observed changes.
Reviewer 4 Report
Comments and Suggestions for Authors
Thank you for the opportunity to review this manuscript. The authors discuss an important topic of declining physical activity during adolescence and a link to loneliness, but I believe this message is hindered by the inclusion of the COVID-19 pandemic and issues around statistical approaches. At this time, I do not believe it is acceptable for publication. My specific concerns are detailed below:
Abstract
· Throughout the paper, differences are discussed without identifying the comparison group. For example, in the abstract, it is stated that physically inactive boys had a higher risk of loneliness, but we don’t know if this is compared to physically active boys or to girls or to some other group.
· The abstract suggests that the COVID-19 pandemic had significant impacts on the trends in loneliness, but this does not seem to bear out in the data or the discussion section of the manuscript.
Introduction
· Ln 55 discusses decreases in physical activity due to social distancing without reporting the possible increase in physical activity during the pandemic that is addressed in the Discussion section. It feels biased and possibly misleading.
Methods
· Figure 1: Please clarify that participation in HUNT1 was not a requirement for follow-up (HUNT2) survey (nor HUNT4 for HUNT COVID) as otherwise is it unclear why the sample size in the two follow-up surveys does not match that of the longitudinal set.
· Physical activity was assessed only in regard to “outside school hours,” even before the pandemic. What was the rationale for limiting this in the PA question? Is it possible that this creates an under-reporting of daily physical activity in the years when school was in session?
· Specific wording of sport participation and loneliness is slightly different across surveys. This is a concern and at the very least needs to be discussed as a limitation in the study.
· Statistical analysis: Chi Square tests are not appropriate for pre-post comparisons within a repeated sample as it violates the assumption of Independence. McNemar’s test is a better fit for this data. This is a major concern as it impacts many of the results.
Results
· Why are the sample sizes for the Loneliness questions so much smaller than for other items? This is never addressed in the manuscript.
· Figure 2: The values do not seem to add up to 100%, as the value is always higher for Sample 2, regardless of which loneliness option is being expressed.
· Again, comparisons are not clearly articulated throughout the manuscript. Table 2 and 3 are very confusing. It is unclear exactly what comparisons are being made. If loneliness is the outcome, how is one of the Loneliness options also the reference group?
Discussion
· Please remove the roman numerals embedded in the text.
· Much of this section contains run-on sentences and grammatical error.
Comments on the Quality of English Language· Manuscript needs extensive editing for English grammar, sentence structure, punctuation and more.
Author Response
Comment summary: Thank you for the opportunity to review this manuscript. The authors discuss an important topic of declining physical activity during adolescence and a link to loneliness, but I believe this message is hindered by the inclusion of the COVID-19 pandemic and issues around statistical approaches. At this time, I do not believe it is acceptable for publication. My specific concerns are detailed below.
Response: Thank you for taking the time to review this manuscript, and the comments on this study. Please find the detailed responses below and the corresponding revisions/corrections highlighted/in track changes in the re-submitted files.
Comment 1:
Abstract
- Throughout the paper, differences are discussed without identifying the comparison group. For example, in the abstract, it is stated that physically inactive boys had a higher risk of loneliness, but we don’t know if this is compared to physically active boys or to girls or to some other group.
- The abstract suggests that the COVID-19 pandemic had significant impacts on the trends in loneliness, but this does not seem to bear out in the data or the discussion section of the manuscript.
Response 1: Thank you. have filled in the manuscript, clarifying what we are comparing. E.g. in the abstract:
Page 1, line 36-37: Physically inactive boys had a higher risk of loneliness compared to physically active boys at both timepoints in Cohort 2, which was higher than in the control cohort 1.
Similar clarifications have been added throughout the paper.
Response 2: Loneliness showed a significant increase among girls in the pandemic (cohort 2) compared to cohort1 two decades earlier. This is presented clearly in the results, but we have chosen not to focus on this in the discussion, as we prioritize the associations between the exposures and loneliness in the discussion section.
Comment 2:
Introduction
- Ln 55 discusses decreases in physical activity due to social distancing without reporting the possible increase in physical activity during the pandemic that is addressed in the Discussion section. It feels biased and possibly misleading.
Methods
- Figure 1: Please clarify that participation in HUNT1 was not a requirement for follow-up (HUNT2) survey (nor HUNT4 for HUNT COVID) as otherwise is it unclear why the sample size in the two follow-up surveys does not match that of the longitudinal set.
- Physical activity was assessed only in regard to “outside school hours,” even before the pandemic. What was the rationale for limiting this in the PA question? Is it possible that this creates an under-reporting of daily physical activity in the years when school was in session?
- Specific wording of sport participation and loneliness is slightly different across surveys. This is a concern and at the very least needs to be discussed as a limitation in the study.
- Statistical analysis: Chi Square tests are not appropriate for pre-post comparisons within a repeated sample as it violates the assumption of Independence. McNemar’s test is a better fit for this data. This is a major concern as it impacts many of the results.
Response 2:
- Thank you for these five points. First, we have rewritten the text from LN55:
Page 2, line 58-64: The implementation of social distancing was expected to result in everyday physical activity decline and hence negatively impacting daily physical activity in general [8,9]. Linked with pandemic-caused restrictions on leisure time activity commitment one could anticipate a further decline also in organized sports participation. This, and the potential impacts of it is important to explore considering the fact that the overall physical activity levels did generally diminish as age increases [10].
- - Figure 1, thanks you for this point. Both longitudinal data sets are based on two new cross-sectional studies, and the longitudinal sample numbers (n) in the figure is actually the analytical sample, which also appears in the results section. We have now indicated this in the figure text.
- - Physical activity was assessed with a commonly used questionnaire that capture leisure time physical activity. Tou could assume an underreporting if the participants have a lot of physical activity during school hours. But the problem with self-reported physical activity is overreporting. But by dichotomizing the variable, we make the reporting more robust against individual levels of under- or over-reporting.
- - Specific wording in sport participation and loneliness, and differences in different time periods is definitely a source of bias. In our study this is minimized due that our two longitudinal data sets follow the same subjects over the extended period, and they have used the same terminology at both timepoints. The differences in terminology between cohorts is compensated with categorizations removing the differences. Furthermore, we totally agree that loneliness can be prone to recall bias, and the need to consider that loneliness was measured with a single question. However, these weakness is not different between the cohorts and timepoints. Loneliness measured with a single question has been shown to correlate well with larger scales, and a single question has the advantage that it produces fewer missing values ​​in the data set than when using several questions.
We have elaborated on this in the text and included it in strength and limitations.
- Statistical analysis: You are correct regarding comparing proportions between two related groups, and we have now used McNemar’s test to analyze paired nominal data to determine if there are differences between these two related groups (baseline and follow-up). These results are now presented in table 1 and the text.
Comment 3:
Results
- Why are the sample sizes for the Loneliness questions so much smaller than for other items? This is never addressed in the manuscript.
- Figure 2: The values do not seem to add up to 100%, as the value is always higher for Sample 2, regardless of which loneliness option is being expressed.
- Again, comparisons are not clearly articulated throughout the manuscript. Table 2 and 3 are very confusing. It is unclear exactly what comparisons are being made. If loneliness is the outcome, how is one of the Loneliness options also the reference group?
Response 3:
Results: The sample size is not smaller for loneliness. There are only a few missing values in some subgroups. Maybe you misunderstood the numbers on loneliness in table 1, as this is the number of participants that are categorized as “feel lonely”, not the sample size. In YH1, 365 girls reporting “feel lonely”, which is 29% of all girls in YH1. Hope this clarifies.
Figure 2: Thank you for this comment. There were some wrong number in this table, this is corrected and now the sum is 100%.
Table 2 and 3: Loneliness at both baseline and follow up are outcomes, and “not lonely” (at each time points) is the reference group. This allows us to have a reference group with the outcome. Then we compare the odds of loneliness at both timepoints with physical activity at baseline in the exposure variable. The reference group for physical activity would typically be the category with the lowest level of activity (e.g., “inactivity”). This allows us to see how different levels of physical activity at baseline affect the odds of being stable lonely, and changes in loneliness (decreasing and increasing loneliness).
Comment 4:
Discussion
- Please remove the roman numerals embedded in the text.
- Much of this section contains run-on sentences and grammatical error.
Response 4:
Thank you for this comment. We have replaced Roman characters with text.
Page 9, line 310-324:
Firstly, physical activity decreases with increasing age during adolescence in both genders whereas loneliness increases only somewhat. Historically, the levels of physical activity in early adolescence increased considerably across the two decades between the first and the second period. When Cohort 2 reached late adolescence approximately 1 year into the pandemic (spring 2021), their previous higher level of physical activity had fallen almost to the same levels as in late adolescence in Cohort 1.
Secondly, an extensive decline in participation in organized sports from the age of 13-15 to the age of 16-19, was observed in both cohorts.
Thirdly, loneliness is more pronounced among girls in both cohorts, and the pro-portion that report loneliness increases with age in both sexes. However, the prevalence increased significantly more in the pandemic period compared to the previous period two decades earlier.
Fourthly, physical inactivity was shown to be associated with loneliness in both cohorts, hence, independent of the pandemic, and
Fifthly, participation in sport seems to be associated with decrease in loneliness in both time periods among girls.
Reviewer 5 Report
Comments and Suggestions for Authors
GENERAL COMMENTS
· Upon exploring the related literature, I was able to find plenty of studies that explored the association between physical activity and loneliness in children and adolescents prior to and during the pandemic.
Despite the premise of the study could be relevant, several points require further clarification or more attention.
· Should this study differ from prior explorations in presenting a longitudinal trajectory of loneliness (over four years) in a large cohort of adolescents aged 13-19? This was not clear to me.
INTRODUCTION
· The introduction presents background from studies that independently examined physical activity or loneliness during the pandemic but were never exposed to previous work on their interrelatedness, which are largely available in the literature.
· Authors are urged to provide further context and background, in light of the existing literature on the topic, to properly pinpoint the research gap and highlight the novel aspect of this study.
· Also, while there is strong evidence from previous work indicating the association between physical activity and loneliness, the authors should elaborate on the rationale of this investigation.
· A better rationale would have led to an effective articulation of the study objectives, which is currently lacking.
· Pg 3; Ln 99-102.you stated “Since participation in organized activities tends to decrease with age, this could bias the follow-up data during the pandemic when the adolescents were four years older. To distinguish the potential effects of the pandemic from those of aging, a previous 4-year follow-up sample (1995-97 to 2000-01) from the same geographical region was used for comparison”. This statement is confusing, I couldn’t understand what the authors intended to convey. This study examined a specific age group (adolescents from 13-19 years old) who typically exhibit consistent behaviors during this period.
METHODS
· Pg 4; Ln 138-139. Provide proper credit to the source of information on which the dichotomization cutoff of active/inactive adolescents was considered.
· Pg 5; Ln 184-200. I suggest placing the sociodemographic factors “Measures” section as the first subheading, prior to discussing the assessment of other variables.
RESULTS
· Results were presented clearly and organized effectively, making them easy to follow in relation to the study objectives.
DISCUSSION
· The conclusion reiterates findings that are well-established in the literature regarding the protective benefits of physical activity and sports participation against loneliness. While the call for public health initiatives to enhance physical activity among adolescents is important, the article would benefit from a more detailed discussion on how these recommendations differ from or build upon previous studies.
· Consider incorporating new insights or data that highlight unique aspects of your approach or suggest innovative strategies to effectively address the issue of loneliness among adolescents.
Author Response
General Comment:
- Upon exploring the related literature, I was able to find plenty of studies that explored the association between physical activity and loneliness in children and adolescents prior to and during the pandemic.
Despite the premise of the study could be relevant, several points require further clarification or more attention.
- Should this study differ from prior explorations in presenting a longitudinal trajectory of loneliness (over four years) in a large cohort of adolescents aged 13-19? This was not clear to me.
Response:
Thank you for your general comment. We have rewritten some of the text, for more clarity that we actually that combine all of you three point in the same study, concerning 1) associations prior and during the pandemic, 2) longitudinal pattern of loneliness an physical activity 3) or is the trends related to increasing age.
INTRODUCTION
Comment 1: · The introduction presents background from studies that independently examined physical activity or loneliness during the pandemic but were never exposed to previous work on their interrelatedness, which are largely available in the literature.
Response 1: Thank you. Yes, but these studies provide valuable insights into how each factor was affected by the pandemic, but as you pointed out, they did not explore the relationship between physical activity and loneliness together. We have added some literature.
Page 2, line 79-82:
A recent study suggests that physical activity is generally inversely related to feelings of loneliness. This relationship varied between pre-pandemic and during-pandemic assessments, with a significant relationship observed only in individuals assessed during the pandemic.
Comment 2: · Authors are urged to provide further context and background, in light of the existing literature on the topic, to properly pinpoint the research gap and highlight the novel aspect of this study.
Response 2: Thank you, provide further context and some new literature in the context in the background.
Comment 3: · Also, while there is strong evidence from previous work indicating the association between physical activity and loneliness, the authors should elaborate on the rationale of this investigation.
Response 3: Ref. above comment, we have elaborated this in the background.
Comment 4: · A better rationale would have led to an effective articulation of the study objectives, which is currently lacking.
Response 4: We have tried to respond to this point, by including new text.
Page 3, line 106-111: Since physical activity and participation in organized activities tends to decrease with age, this could bias follow-up data during the pandemic when the adolescents were four years older. To distinguish the potential effects of the pandemic from those of aging, a previous 4-year follow-up sample from the Trøndelag health Study (The HUNT study) in1995-97 to 2000-01 used for comparison
Comment 5: · Pg 3; Ln 99-102.you stated “Since participation in organized activities tends to decrease with age, this could bias the follow-up data during the pandemic when the adolescents were four years older. To distinguish the potential effects of the pandemic from those of aging, a previous 4-year follow-up sample (1995-97 to 2000-01) from the same geographical region was used for comparison”. This statement is confusing, I couldn’t understand what the authors intended to convey. This study examined a specific age group (adolescents from 13-19 years old) who typically exhibit consistent behaviors during this period.
Response 5: Thank you, this has been removed from the text, and we see that it only became confusing and not significant for the study.
METHODS
Comment 6: · Pg 4; Ln 138-139. Provide proper credit to the source of information on which the dichotomization cutoff of active/inactive adolescents was considered.
Response 6: There has been a formatting error here, this has been corrected
Comment 7: · Pg 5; Ln 184-200. I suggest placing the sociodemographic factors “Measures” section as the first subheading, prior to discussing the assessment of other variables.
Response 7: We agree and have placing SES as first subheading.
RESULTS
Comment 8: · Results were presented clearly and organized effectively, making them easy to follow in relation to the study objectives.
Response 8: Thank you.
DISCUSSION
Comment 9: · The conclusion reiterates findings that are well-established in the literature regarding the protective benefits of physical activity and sports participation against loneliness. While the call for public health initiatives to enhance physical activity among adolescents is important, the article would benefit from a more detailed discussion on how these recommendations differ from or build upon previous studies.
Response 9: Thank you for this comment. We have elaborated a little more in the Conclusion on how our findings complement previous studies.
Page 11, line 510-521: adds a nuanced understanding of how these activities impact different genders. Considering the negative trend and impacts of loneliness, and the protective role of physical activity against rising loneliness during adolescence for both genders, with sports participation linked to reduced loneliness in girls and a lower risk of loneliness in boys, it is advised that public health initiatives focus on boosting physical activity and minimizing sports drop-out rates among adolescents. A special focus should be on monitoring post lock-down situations; however, the biggest challenge is linked to the impact of drop-out due to increasing age, and a much greater focus must be placed on preventing this in adolescence. The added emphasis on monitoring post-lockdown situations and addressing the challenges of age-related drop-out provides a more comprehensive approach to sustaining adolescent engagement in physical activities.
Comment 10: · Consider incorporating new insights or data that highlight unique aspects of your approach or suggest innovative strategies to effectively address the issue of loneliness among adolescents.
Response 10: Thanks for this comment and this is a very important area. I have not included this in the manuscript, but believe it is important incorporating strategies that align with the Act-Belong-Commit (ABC) framework, act (do something), belong (do something with Somone), commit (do something meaningful).
Round 2
Reviewer 1 Report
Comments and Suggestions for Authors
Dear authors
Thank you for your thorough response and revisions. There is one issue where I disagree with you, the choice of confounder variables (mental distress, self-rated health). I accept that the chosen variables can influence loneliness but this is not a sufficient reason to include them as confounders. The chosen variables may be influenced by physical activity in which case they are not confounders but mediators. You MUST adjust for confounders but you MUST NOT control for mediators.
I strongly suggest that you A) either drop these variables as confounders in the statistical analysis or B) provide a convincing justification that they are not a result of physical activity but a precursor of physical activity.
Author Response
Comment from reviewer: Thank you for your thorough response and revisions. There is one issue where I disagree with you, the choice of confounder variables (mental distress, self-rated health). I accept that the chosen variables can influence loneliness but this is not a sufficient reason to include them as confounders. The chosen variables may be influenced by physical activity in which case they are not confounders but mediators. You MUST adjust for confounders but you MUST NOT control for mediators.
I strongly suggest that you A) either drop these variables as confounders in the statistical analysis or B) provide a convincing justification that they are not a result of physical activity but a precursor of physical activity.
Response: Thank you again for this point. We agree that the health variables (mental distress, self-rated health) can act as mediators between physical activity and loneliness (which we also argue for). However, they can also affect physical activity. This means that if we use directed acyclic graphs (DAGs), the arrow will go both ways between physical activity and health (mental distress, self-rated health), and it is in this context that we recommend, and is the rationale for our choices, to handle these variables as confounders.
This is an interesting discussion, an probably not any quick answer to handle this, but we have re-run the analyses, removing the two health variables, and therefore not controlling for these in the “new” model 2 (Table 2 and 3). This was your first suggest. Additionally, we chose to include supplementary tables and include all three models in these supplementary tables (Table S1 and Table S2), where we show how associations appear when we treat mental distress and self-rated health as confounding factors. I think that is fair for the reader, and we also see that this does not affect our findings and conclusions.
We have also tested for multicollinearity, and the VIF is below 2 for all variables.
Reviewer 3 Report
Comments and Suggestions for Authors
Dear authors.
Having seriously reviewed your replies, I regret to say that I am not convinced by them. The research has a number of implicit problems with the material and method. The lack of reliable instruments makes the data unreliable.
Author Response
Comment summary: Dear authors.
Having seriously reviewed your replies, I regret to say that I am not convinced by them. The research has a number of implicit problems with the material and method. The lack of reliable instruments makes the data unreliable.
Response: Thank you again, for taking you time. But the argument regarding reliable instrument and that the results are outdating by other publications are not valid. Also ref. to academic editors notes.
Reviewer 4 Report
Comments and Suggestions for Authors
I appreciate the authors’ efforts to correct the statistical approaches used in this manuscript and the improvement of some of the tables and figures. However, I still have large concerns that the data is being overinterpreted and discussed only in ways that support the authors’ hypotheses rather than following the data.
Materials and Methods
· Some sample sizes in the text do not match Figure 1
Discussion
· Ln 314-315: Statement that the higher level of PA had fallen to “almost the same levels” in Cohort 2 compared to Cohort 1 – But PA was still higher in Cohort 2 (even despite the pandemic) and no statistical comparison was made between either the two post-measurements in each cohort or the change over time between the two cohorts. Be careful of over-interpreting.
· Ln 316-317: Ignores the (again, not statistically tested) fact that sport participation was higher at both the first and second measurement in the second cohort compared to the first.
· Ln 324-325: This effect was in opposite directions between the two cohorts but this is never discussed (OR 0.66 in Cohort 1; OR 1.85 in Cohort 2)
· Ln 335: States that the COVID pandemic was associated with less physical activity. What test is this based on? There is no statistical comparison to base this claim on. The decrease in PA in Cohort 2 could simply be regression to the mean combined with decrease in age.
· Ln 367: Was any test of the relationship between sport participation and physical activity level actually conducted to support this assumption?
· Ln 389 states that a strength of the study is consistent methods and instruments even though you say yourself in Ln 408 that measurements were non-identical.
· Ln 413-414: The ages and grades do not seem to correspond. Did they get switched?
· Ln 391: American YRBSS asks about sport team participation, which seems just as “comprehensive” as the one-question approach to measuring sport participation used here.
Comments on the Quality of English LanguageSome minor errors such as noun/verb agreement and sentence structure still exist, but has been greatly improved.
Author Response
Comment summary: I appreciate the authors’ efforts to correct the statistical approaches used in this manuscript and the improvement of some of the tables and figures. However, I still have large concerns that the data is being overinterpreted and discussed only in ways that support the authors’ hypotheses rather than following the data.
Response: Thank you for taking the time to review this manuscript, and the comments on this study. Please find the detailed responses below and the corresponding revisions/corrections changes in the re-submitted files.
Materials and Methods
- Some sample sizes in the text do not match Figure 1
Response: Sorry, but cannot recover that there is a mismatch between text and figure 1.
Discussion
- Ln 314-315: Statement that the higher level of PA had fallen to “almost the same levels” in Cohort 2 compared to Cohort 1 – But PA was still higher in Cohort 2 (even despite the pandemic) and no statistical comparison was made between either the two post-measurements in each cohort or the change over time between the two cohorts. Be careful of over-interpreting.
Response: Page 9, line 315-317, rephrased: When Cohort 2 reached late adolescence approximately 1 year into the pandemic (spring 2021), their previous higher level of physical activity significantly decreased.
- Ln 316-317: Ignores the (again, not statistically tested) fact that sport participation was higher at both the first and second measurement in the second cohort compared to the first.
Response: Yes, but we have tested for changes into the two cohorts, and in both here are significantly differences in the proportion in both cohorts. But as you point out, we have not tested between the two cohorts.
- Ln 324-325: This effect was in opposite directions between the two cohorts but this is never discussed (OR 0.66 in Cohort 1; OR 1.85 in Cohort 2)
Response: That is correct. In the fully adjusted model, this was clearly evident. But the OR for several of the loneliness outcomes points in one direction and therefore we express that it "seems" that participation in sports is associated with….., in the manuscript. But agree that the findings are more weakly linked to sports participation compared to physical activity.
- Ln 335: States that the COVID pandemic was associated with less physical activity. What test is this based on? There is no statistical comparison to base this claim on. The decrease in PA in Cohort 2 could simply be regression to the mean combined with decrease in age.
Response: Agree, this could be in combination with decrease age. Therefore rephrased the sentence.
Page 10, line 336-338: Analyzing our combined study sample showed that the COVID-19 pandemic was associated with a decrease in physical activity compared to three years earlier (before the pandemic).
- Ln 367: Was any test of the relationship between sport participation and physical activity level actually conducted to support this assumption?
Response: No tests, only the evidence supporting the assumption that physical activity can reduce loneliness, and sport participation in sports can lead increased physical activity.
- Ln 389 states that a strength of the study is consistent methods and instruments even though you say yourself in Ln 408 that measurements were non-identical.
Response: This is two folded. In the longitudinal design (each cohort) we have identical instruments, but some gaps are made between the cohorts. Most of it is to adapt the social development related to e.g. term use.
- Ln 413-414: The ages and grades do not seem to correspond. Did they get switched?
Response: Thank you, it’s a typo, they've been switched
- Ln 391: American YRBSS asks about sport team participation, which seems just as “comprehensive” as the one-question approach to measuring sport participation used here.
Response: Yes, YRBSS does indeed include questions about sports team participation (including those run by their school or community groups), and provide a broad measure of sports participation, and may seem very comprehensive as in our data. It might seem similar to single-question approach, but it allow a more nuanced understanding of how sports participation correlates with aspects of youth health and behavior.
Some minor errors such as noun/verb agreement and sentence structure still exist, but has been greatly improved.
Response: Thank you, we have tried to look at the sentence structure and some errors was found.
Reviewer 5 Report
Comments and Suggestions for Authors
Authors have addressed previous concerns. I have no further comments
Author Response
General Comment:
Authors have addressed previous concerns. I have no further comments
Response:
Thank you very much for taking the time to review this manuscript. Your comments really helped us to improve the manuscript.